# Effect of Cr Addition on Magnetic Properties and Corrosion Resistance of Optimized Co and Fe-Based Amorphous Alloys

Jonghee Han [1], Jihyun Hong [2], Seoyeon Kwon [1] and Haein Choi-Yim [1,*]

[1] Department of Physics, Sookmyung Women's University, Seoul 04310, Korea; jonghee6150@sookmyung.ac.kr (J.H.); sykwon39@sookmyung.ac.kr (S.K.)
[2] Center for Energy Materials Research, Korea Institute of Science and Technology, Seoul 02792, Korea; jihyunh@kist.re.kr
[*] Correspondence: haein@sookmyung.ac.kr

**Abstract:** In the Fe-Co alloy system, the addition of Cr improves the glass-forming ability (GFA) with superior soft magnetic properties such as high saturation magnetization ($M_s$) and low coercivity ($H_c$). In addition, Cr is considered to be an important factor for improving the corrosion resistance of Fe-based amorphous alloy. Therefore, in the present study, we investigated the GFA, soft magnetic properties, and corrosion resistance of the as-spun ribbons in $[Co_{0.075}Fe_{0.675}B_{0.2}Si_{0.05}]_{100-x}Cr_x$ ($x$ = 0–8) alloy system. The ribbons were produced using the melt-spinning technique and were characterized by X-ray diffraction, differential scanning calorimetry, thermomechanical analysis, and vibrating sample magnetometer. The Co-Fe-B-Si-Cr alloys exhibited high thermal stability and a high $M_s$ of 0.93–1.53 T. Corrosion properties were evaluated by cyclic voltammetry. The addition of Cr improved the corrosion resistance of the alloys. The alloys with a higher Cr content exhibited a higher corrosion resistance. The optimum combination with soft magnetic properties and corrosion resistance of Fe-co based amorphous alloys can be utilized for extensive fields of application through a variation of Cr contents.

**Keywords:** amorphous; magnetic properties; GFA; corrosion resistance

## 1. Introduction

Fe-based amorphous alloys are very attractive for industrial application due to the possibility of obtaining alloys with superior mechanical properties such as high strength, good soft magnetic properties and have been used in application fields of magnetic and high specific strength materials [1–3]. In addition, these alloys are considered to be excellent candidates for protecting steel surfaces because they have a high crystallization temperature ($T_x$), as well as high corrosion and wear resistance [4].

However, due to the relatively low glass-forming ability (GFA), there is a limit for multifunctional applications of Fe-based amorphous alloys [5–8]. Therefore, many studies have been conducted on various compositions by adding several elements to improve GFA with high corrosion resistance and good soft magnetic properties of the Fe-based amorphous alloys for commercial applications [1,9–11]. For instance, Fe-Co-B-Si amorphous alloys named METGLAS were first developed to exhibit good soft magnetic properties in 1974 [12]. The Fe-Co alloy system showed a high saturation magnetization ($M_s$) and permeability as compared to Fe-only and Co-only systems [13]. Also, the substitution of Co for Fe was found to enhance the GFA [14].

Furthermore, some Fe-based compositions with Cr for high corrosion resistance, including Fe-Cr-Mo-C-B [2], Fe-Co-Si-B-Nb-Cr [4], Fe-Cr-Mo-Y-C-B [15], have been developed. Cr is a major element in increasing the corrosion resistance of Fe-based amorphous alloy [16]. Therefore, the Cr content in the Fe-Co based alloy can be essential to improve the soft magnetic properties and corrosion resistance together with a high GFA [17,18].

For a practical application of the Fe-Co based amorphous alloys in severe environments, optimization of corrosion resistance, GFA, and soft magnetic properties is needed [8].

In a previous study, the amorphous ribbon with $Co_{7.2}Fe_{64.8}B_{19.2}Si_{4.8}Cr_4$ composition among the $[Co_{1-x}Fe_x]_{72}B_{19.2}Si_{4.8}Cr_4$ $(0 \leq x \leq 1)$ alloys showed superior soft magnetic properties such as high $M_s$ and low coercivity $(H_c)$ [18]. Accordingly, in the present study, we focused on the $Co_{7.2}Fe_{64.8}B_{19.2}Si_{4.8}Cr_4$ and identified the improving effects of Cr variation on GFA, corrosion resistance as well as soft magnetic properties.

## 2. Materials and Methods

Multicomponent alloy ingots with a composition $[Co_{0.075}Fe_{0.675}B_{0.2}Si_{0.05}]_{100-x}Cr_x$ $(x = 0–8)$ were prepared by arc melting mixtures of high purity Co (99.95%), Fe (99.95%), Si (99.99%), Cr (99.99%), and industry-grade pre-alloys of FeB (99%) under Ti-gettered argon atmosphere and then re-melted at least four times to achieve homogeneity. Amorphous ribbons of each composition were produced using the melt spinning technique in an argon atmosphere with a wheel speed of 39.2 m/s. The width and thickness of ribbons were 2mm and 20~30μm, respectively. Phase structures of as-spun ribbons were identified by X-ray diffraction (XRD, D8 Advance, Bruker, Karlsruhe, Germany) with Cu-Kα radiation. Thermal stability associated, crystallization temperature $(T_x)$, with glass transition temperature $(T_g)$ and supercooled liquid region $(\Delta T_x = T_x - T_g)$ were investigated by a differential scanning calorimetry (DSC, Labsys N-650, SCINCO, SEOUL, Korea) at the heating rate of 0.34 °C/s under an argon flow and a thermomechanical analysis (TMA, Q400, TA Instruments, New Castle, DE, USA) in the tensile mode. TMA measurement was conducted with the heating rate 0.34 °C/s, a force of 1.2 N, and an initial height of 8 mm. The values of $M_s$ were evaluated with a vibrating sample magnetometer (VSM, EV9, MicroSense, Lowell, MA, USA) at room temperature in the maximum applied field $(H_m)$ of 10 kOe. Additionally, the density of the specimens was determined using a helium pycnometer (AccuPyc II, Micromeritics, USA).

The corrosion behavior of the alloys was investigated by electrochemical measurements in 0.1 M HCl solution open to air at 298 K. The electrochemical measurement was conducted using a potentiostat (SP-200, Biologic, France) in a three-electrode cell using a graphite counter electrode and a saturated calomel electrode (SCE) as the reference electrode. The working electrode was the as-spun ribbon, with an immersed average area 0.8 cm². Cyclic voltammograms were measured with the potential scan rate of 1.7 mV within the range of −750 mV to 150 mV (vs SCE), after the open-circuit potentials had been stabilized for 20 min. Five cycles of anodic and cathodic scans were repeated unless the ribbons were ruptured.

## 3. Results & Discussion

First, we measured XRD patterns to confirm that all specimens had an amorphous structure. Figure 1 shows that the XRD patterns of the as-spun $[Co_{0.075}Fe_{0.675}B_{0.2}Si_{0.05}]_{100-x}Cr_x$ $(x = 0–8)$ alloys. The patterns consisted only of broad halo humps without any sharp diffraction peak of crystalline phases, indicating that the specimens had a fully amorphous structure.

To investigate the thermal properties of as-spun $[Co_{0.075}Fe_{0.675}B_{0.2}Si_{0.05}]_{100-x}Cr_x$ $(x = 0–8)$ ribbons such as $T_{x,DSC}$, we obtained DSC curves as shown in Figure 2a,b. Also, as shown in Table 1, all of the onset $T_{x,DSC}$ values were marked with the arrows in Figure 2. For all samples, DSC curves showed a sharp exothermic peak near 550 °C. We attributed the peak to the single-stage crystallization of as-spun $[Co_{0.075}Fe_{0.675}B_{0.2}Si_{0.05}]_{100-x}Cr_x$ $(x = 0–3)$ amorphous ribbons and the fine second exothermic peak that implied that a two-stage crystallization process was observed in DSC curves of $[Co_{0.075}Fe_{0.675}B_{0.2}Si_{0.05}]_{100-x}Cr_x$ $(x = 4, 5, 7, 8)$ amorphous ribbons. In all specimens, the position of the first exothermic peak did not change significantly within the range of 546 °C–548 °C as the Cr content increased. As a higher $T_{x,DSC}$ signifies higher thermal stability of the ribbon, the thermal stability of the as-spun $[Co_{0.075}Fe_{0.675}B_{0.2}Si_{0.05}]_{100-x}Cr_x$ $(x = 0–8)$ ribbons were consistent [19] and indicated independence on the addition of Cr. In the DSC curves, the glass transition tem-

perature ($T_g$) was not shown exactly in any ribbon. In general, no glass transition through the DSC experiment was observed for Fe-rich amorphous alloys [20]. In addition, Cr addition tended to decrease the GFA, since the liquid's temperature significantly increases with an increase of Cr content [8]. Thus it can be speculated that the GFAs of these alloys were marginal [14].

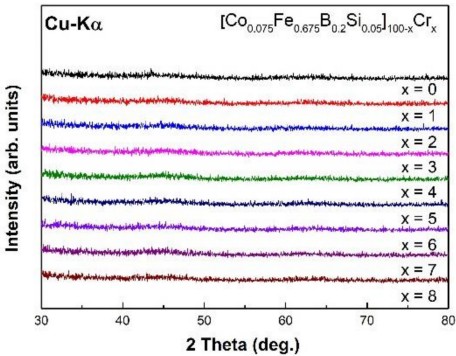

**Figure 1.** The X-ray diffraction (XRD) patterns of $[Co_{0.075}Fe_{0.675}B_{0.2}Si_{0.05}]_{100-x}Cr_x$ ($x$ = 0–8) as-spun ribbons.

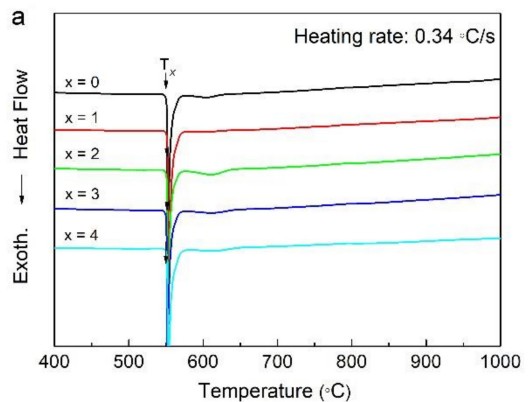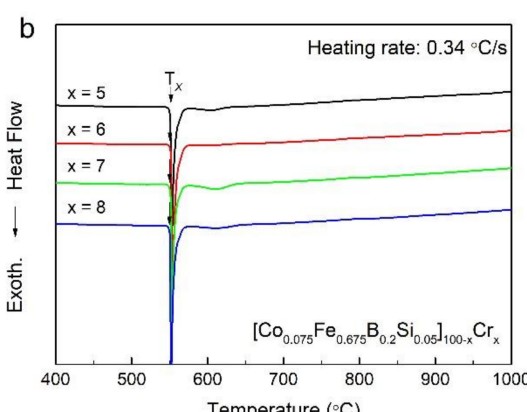

**Figure 2.** The differential scanning calorimetry (DSC) patterns of (**a**) $[Co_{0.075}Fe_{0.675}B_{0.2}Si_{0.05}]_{100-x}Cr_x$ ($x$ = 0–4) as-spun ribbons, and (**b**) $[Co_{0.075}Fe_{0.675}B_{0.2}Si_{0.05}]_{100-x}Cr_x$ ($x$ = 5–8) as-spun ribbons.

**Table 1.** Summary of crystallization temperatures ($T_x$) values measured by DSC.

| $x$ | $[Co_{0.075}Fe_{0.675}B_{0.2}Si_{0.05}]_{100-x}Cr_x$ | $T_{x,DSC}$ (°C) |
|---|---|---|
| 0 | $[Co_{0.075}Fe_{0.675}B_{0.2}Si_{0.05}]_{100}$ | 546.4 |
| 1 | $[Co_{0.075}Fe_{0.675}B_{0.2}Si_{0.05}]_{99}Cr_1$ | 547.7 |
| 2 | $[Co_{0.075}Fe_{0.675}B_{0.2}Si_{0.05}]_{98}Cr_2$ | 548.3 |
| 3 | $[Co_{0.075}Fe_{0.675}B_{0.2}Si_{0.05}]_{97}Cr_3$ | 546.8 |
| 4 | $[Co_{0.075}Fe_{0.675}B_{0.2}Si_{0.05}]_{96}Cr_4$ | 546.8 |
| 5 | $[Co_{0.075}Fe_{0.675}B_{0.2}Si_{0.05}]_{95}Cr_5$ | 548.2 |
| 6 | $[Co_{0.075}Fe_{0.675}B_{0.2}Si_{0.05}]_{94}Cr_6$ | 548.7 |
| 7 | $[Co_{0.075}Fe_{0.675}B_{0.2}Si_{0.05}]_{93}Cr_7$ | 548.2 |
| 8 | $[Co_{0.075}Fe_{0.675}B_{0.2}Si_{0.05}]_{92}Cr_8$ | 547.4 |

In order to further confirm that the endothermic reaction before crystallization was due to the glass transition [21], the Co-Fe-B-Si-Cr amorphous alloys were investigated by TMA. The results of the TMA experiment are presented as a dimension change versus temperature plot, which are shown in Figure 3. The dimension change of the Co-Fe-B-Si-Cr alloys begins to increase around 520 °C and the increase in dimension change in the temperature range between 520 and 550 °C indicates that this range is the supercooled region

($\Delta T_x = T_{x,\text{TMA}} - T_g$) [21]. In the plots of Figure 3, the values of $T_{x,\text{TMA}}$ was determined as the point at which thermal expansion occurs according to crystallization [22]. In addition, the values of $T_g$ were determined from the intersection of two tangential lines. $T_{x,\text{TMA}}$, and $T_g$ are marked with the arrows in Figure 3a,b. These thermal properties measured by TMA are summarized in Table 2. As can be seen in Table 2, $T_{x,\text{TMA}}$ values almost correspond to $T_{x,\text{DSC}}$.

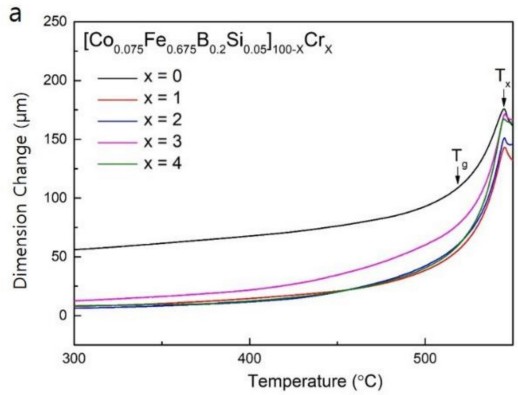 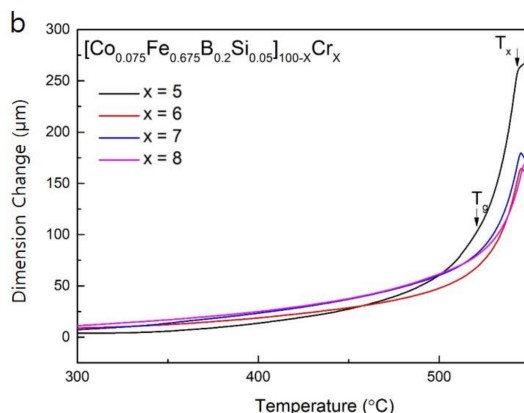

**Figure 3.** Temperature dependence of the dimension change for (**a**) $[Co_{0.075}Fe_{0.675}B_{0.2}Si_{0.05}]_{100-x}Cr_x$ ($x$ = 0–4) as-spun ribbons, and (**b**) $[Co_{0.075}Fe_{0.675}B_{0.2}Si_{0.05}]_{100-x}Cr_x$ ($x$ = 5–8) as-spun ribbons.

**Table 2.** Summary of thermal properties measured by TMA.

| $x$ | $[Co_{0.075}Fe_{0.675}B_{0.2}Si_{0.05}]_{100-x}Cr_x$ | $T_{x,\text{TMA}}$ (°C) | $T_g$ (°C) | $\Delta T_x$ (°C) |
|---|---|---|---|---|
| 0 | $[Co_{0.075}Fe_{0.675}B_{0.2}Si_{0.05}]_{100}$ | 544.92 | 518.06 | 26.86 |
| 1 | $[Co_{0.075}Fe_{0.675}B_{0.2}Si_{0.05}]_{99}Cr_1$ | 545.06 | 522.65 | 22.41 |
| 2 | $[Co_{0.075}Fe_{0.675}B_{0.2}Si_{0.05}]_{98}Cr_2$ | 544.97 | 523.38 | 21.59 |
| 3 | $[Co_{0.075}Fe_{0.675}B_{0.2}Si_{0.05}]_{97}Cr_3$ | 545.09 | 523.03 | 22.06 |
| 4 | $[Co_{0.075}Fe_{0.675}B_{0.2}Si_{0.05}]_{96}Cr_4$ | 544.19 | 524.68 | 19.51 |
| 5 | $[Co_{0.075}Fe_{0.675}B_{0.2}Si_{0.05}]_{95}Cr_5$ | 543.23 | 521.91 | 21.32 |
| 6 | $[Co_{0.075}Fe_{0.675}B_{0.2}Si_{0.05}]_{94}Cr_6$ | 544.86 | 524.39 | 20.47 |
| 7 | $[Co_{0.075}Fe_{0.675}B_{0.2}Si_{0.05}]_{93}Cr_7$ | 544.76 | 522.46 | 22.30 |
| 8 | $[Co_{0.075}Fe_{0.675}B_{0.2}Si_{0.05}]_{92}Cr_8$ | 546.77 | 524.25 | 22.52 |

The hysteresis M-H loops measured at room temperature of the $[Co_{0.075}Fe_{0.675}B_{0.2}Si_{0.05}]_{100-x}Cr_x$ ($x$ = 0–8) amorphous ribbons are shown in Figure 4a,b. All the loops were typical soft magnetic loops with a high $M_s$ and a low $H_c$. As shown in Figure 4, the addition of Cr caused a decrease in $M_s$ for the as-spun ribbons. With an increase of the Cr content from 0 to 8 at.%, the values of $M_s$ gradually decreased from 1.53 to 0.93 T due to the reduction of the content of Fe by Cr element [22]. However, the amorphous ribbons with less than 5 at.% Cr content all showed high $M_s$ over 1.2 T. The highest $M_s$ value of $[Co_{0.075}Fe_{0.675}B_{0.2}Si_{0.05}]_{100}$ was relatively high as compared to those of the Fe-based amorphous alloys such as FeBSiP, FeCrMoPBC [20,23]. The detail values of $M_s$ for all specimens are summarized in Table 3.

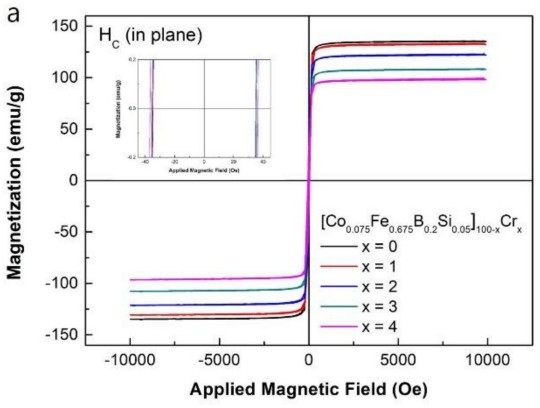
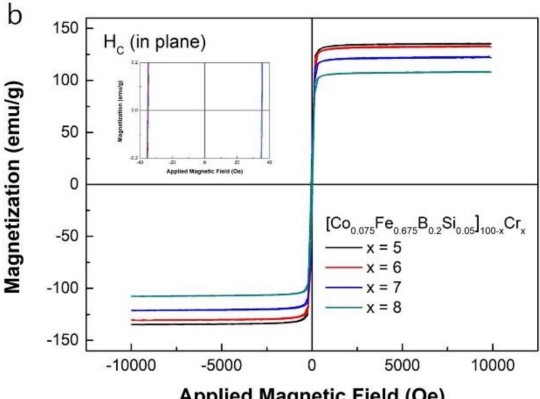

**Figure 4.** Hysteresis M-H loop of (**a**) the as-spun $[Co_{0.075}Fe_{0.675}B_{0.2}Si_{0.05}]_{100-x}Cr_x$ ($x$ = 0–4), and (**b**) the as-spun $[Co_{0.075}Fe_{0.675}B_{0.2}Si_{0.05}]_{100-x}Cr_x$ ($x$ = 5–8).

**Table 3.** Summary of density ($\rho$), saturation magnetization ($M_s$) values.

| x | $[Co_{0.075}Fe_{0.675}B_{0.2}Si_{0.05}]_{100-x}Cr_x$ | $\rho$ (g/cm³) | $M_s$ | |
|---|---|---|---|---|
| | | | emu/g | T |
| 0 | $[Co_{0.075}Fe_{0.675}B_{0.2}Si_{0.05}]_{100}$ | 6.69 | 182.09 | 1.53 |
| 1 | $[Co_{0.075}Fe_{0.675}B_{0.2}Si_{0.05}]_{99}Cr_1$ | 6.70 | 168.09 | 1.42 |
| 2 | $[Co_{0.075}Fe_{0.675}B_{0.2}Si_{0.05}]_{98}Cr_2$ | 6.70 | 159.29 | 1.34 |
| 3 | $[Co_{0.075}Fe_{0.675}B_{0.2}Si_{0.05}]_{97}Cr_3$ | 6.71 | 153.89 | 1.30 |
| 4 | $[Co_{0.075}Fe_{0.675}B_{0.2}Si_{0.05}]_{96}Cr_4$ | 6.71 | 148.97 | 1.26 |
| 5 | $[Co_{0.075}Fe_{0.675}B_{0.2}Si_{0.05}]_{95}Cr_5$ | 6.72 | 137.72 | 1.16 |
| 6 | $[Co_{0.075}Fe_{0.675}B_{0.2}Si_{0.05}]_{94}Cr_6$ | 6.72 | 133.54 | 1.13 |
| 7 | $[Co_{0.075}Fe_{0.675}B_{0.2}Si_{0.05}]_{93}Cr_7$ | 6.73 | 121.12 | 1.02 |
| 8 | $[Co_{0.075}Fe_{0.675}B_{0.2}Si_{0.05}]_{92}Cr_8$ | 6.73 | 110.23 | 0.93 |

To understand the effect of Cr addition on corrosion behavior, we recorded cyclic voltammograms of the alloys in an acidic 0.1 M HCl solution. As shown in Figure 5, the degree of Cr addition significantly altered the current responses of the alloy specimens. Overall, the average current density became lower as the Cr contents increased, confirming that the addition of Cr improved the corrosion resistance of the alloys. For the alloys with $x < 6$, the current density continuously increased until the potential reached 0.15 V cutoff condition during the anodic scan and then decreased during the cathodic scan. We noted the ruptures of ribbons with a low Cr content ($x$ = 0, 3, and 4) occurring in the middle of the cyclic voltammetry (CV) measurements (indicated with arrows). The disconnection of the ribbons as the result of severe corrosion caused a sudden current drop. In sharp contrast, the alloys with $x$ = 6, 7, and 8 exhibited reproducible current responses for five CV cycles without suffering from the failure, enduring for a longer duration. The result confirms the positive effect of Cr doping on anti-corrosion properties. For the Cr-rich alloys, the current densities did not continuously increase during the anodic scan. Instead, there was a decay after the initial increase in current density, as displayed in Figure 4b. We attributed the origin of the decrease in current density to the formation of the passivation layer on the surface of the high-Cr alloys during the anodic scan, which suppressed further proceeding of corrosion at a high potential [24].

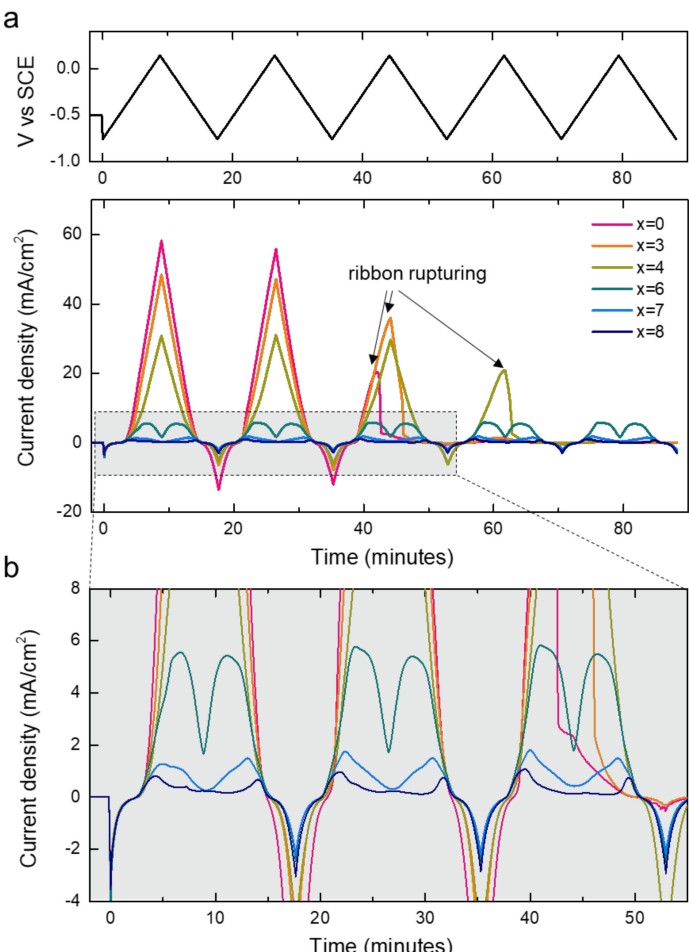

**Figure 5.** (**a**) Cyclic voltammograms of $[Co_{0.075}Fe_{0.675}B_{0.2}Si_{0.05}]_{100-x}Cr_x$ ($x$ = 0, 3, 4, 6, 7, 8) in 0.1 M HCl recorded within −0.75 V to 0.15 V vs SCE for five electrochemical cycles. Top panel shows voltage controlled during the CV measurements. (**b**) Magnified cyclic voltammograms in the regime highlighted with a dashed rectangle in (**a**).

In Figure 6, we further compare the polarization diagrams of the alloys that were re-plotted from the CV curves of the first two anodic scans. With the corrosion potentials ($E_{corr}$) of all the alloys recorded within the potential range of 12 mV (−0.496 V to −0.508 V) for the initial CV cycle, we found the $E_{corr}$ of alloys wider distributed within a potential range of 60 mV (−0.450 V to −0.510 V) for the second cycle. The $E_{corr}$ of alloys with a lower Cr content changed more as compared to those with a higher Cr content ($\Delta E_{corr}$: $x$ = 0 > 3 ≈ 4 > 6 ≈ 7 ≈ 8), as shown in Figure 6b. The composition of the alloy surface can change if an alloy component dissolves preferentially and more rigorously than others, which eventually causes irreversible shifts in $E_{corr}$. Interestingly, the cathodic current density at the potential below $E_{corr}$ also significantly increased for the alloys with a low Cr content (highlighted with grey arrows in Figure 6b). This result supports the hypothesis about the formation of the passivation layer of high-Cr alloys and confirms the anti-corrosion effect of Cr in the Fe-based amorphous alloys.

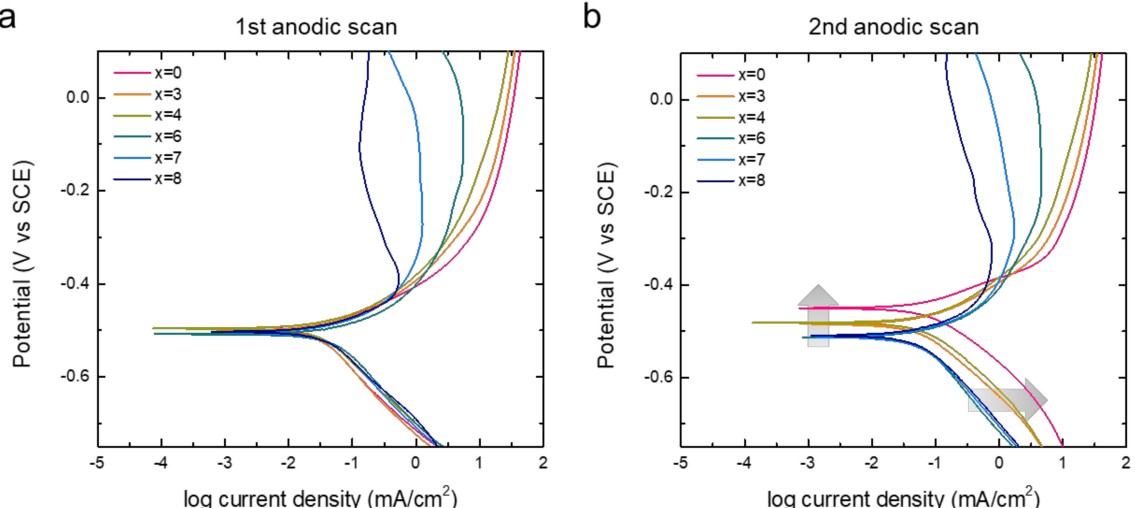

**Figure 6.** Polarization diagrams of $[Co_{0.075}Fe_{0.675}B_{0.2}Si_{0.05}]_{100-x}Cr_x$ ($x$ = 0, 3, 4, 6, 7, 8) measured in Figure 5. Panels (**a**,**b**) show the diagrams for first and second anodic scans, respectively.

## 4. Conclusions

This study investigated the effects of the Cr addition on GFA, magnetic properties, and corrosion resistance of as-spun $[Co_{0.075}Fe_{0.675}B_{0.2}Si_{0.05}]_{100-x}Cr_x$ ($x$ = 0–8) ribbons. Although the GFA of the amorphous alloys was relatively low because of the Cr addition, we found it to be effective, as the Cr addition enhanced the corrosion resistance with a high $M_s$ of 0.93–1.53 T.

In the 0.1 M HCl solution, the anodic current density during the CV measurement decreased from about 60 to 0.4 mA/cm$^2$ as the Cr contents increased. The significant reduction of the current density confirms that the addition of Cr improves the corrosion resistance of the alloys. Especially for the alloys with $x$ > 4, due to the passivation behavior, the ribbons with a higher Cr content ($x$ = 6, 7, and 8) exhibit reproducible current responses for five CV cycles without rupturing due to the formation of a passive layer. In contrast, the alloys with a low Cr content ($x$ = 0, 3, and 4) showed a significant increase in current density and $E_{corr}$ after the first CV cycle as the result of severe corrosion. The results prove the anti-corrosion effect of Cr in the Fe-based amorphous alloys.

**Author Contributions:** Conceptualization, J.H. (Jonghee Han) and J.H. (Jihyun Hong); methodology, S.K. and J.H. (Jihyun Hong); validation, J.H. (Jonghee Han), J.H. (Jihyun Hong) and S.K.; formal analysis, J.H. (Jonghee Han), J.H. (Jihyun Hong) and S.K; investigation, J.H. (Jonghee Han); resources, S.K.; data curation, J.H. (Jonghee Han) and J.H. (Jihyun Hong); writing–original draft preparation, J.H. (Jonghee Han) and J.H. (Jihyun Hong); writing–review and editing, J.H. (Jonghee Han), J.H. (Jihyun Hong), and H.C.-Y.; visualization, J.H. (Jonghee Han).; project administration, H.C.-Y.; funding acquisition, H.C.-Y. All authors have read and agreed to the published version of the manuscript.

**Funding:** This research was supported by the Basic Science Research Program through the National Research Foundation of Korea (NRF) funded by the Ministry of Science, ICT and Future Planning (MSIP, grant no. 2018006784) and Korea Energy Technology Evaluation and Planning (KETEP, grant no. 20181110100410) of Ministry of Trade, Industry and Energy of Republic of Korea.

**Data Availability Statement:** No new data were created or analyzed in this study. Data sharing is not applicable to this article.

**Conflicts of Interest:** The authors declare no conflict of interest.

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
