# Peer review of "Effect of Cr Addition on Magnetic Properties and Corrosion Resistance of Optimized Co and Fe-Based Amorphous Alloys"

_metals, doi:10.3390/met11020304_

Round 1
Reviewer 1 Report
It is good paper and the presented data are interesting and generally sound. Technically valuable results. Only few manor points for corrections and clarification for minor revision:
- There are at least two mistakes in the language - lines 23 and 67
- It will be good to indicate the errors in the temperatures for crystallization and thermomechanical measurements. I am not convinced that this is the accurate temperature. It is temperature on the instrument, but not the actual ribbon temperature. The authors need to provide convincing data and explanation
Author Response
Dear reviewer,
Thank you for completing the review of our paper in a timely manner. We also thank the reviewer for his/her positive comments on our paper. We have revised the manuscript in response to the comments, highlighting changes in yellow. Our detailed responses to his/her comments are attached.
Please see the attachment.
Once again, we are grateful to you and the reviewer for the great efforts. We look forward to hearing from you about its acceptability.

Reviewer 2 Report
Please see and follow my :sticky notes: in the attached docx file.

Author Response

(The authors gave the same response as above.)

Reviewer 3 Report
This is an excellent paper reporting the effects of Cr addition on the magnetic properties and corrosion resistance of an optimized Fe-Co-B-Si amorphous alloy. The authors performed structural analysis, thermal and themomechanical analysis, magnetic hysteresis and electrochemical measurements on a series of alloy compositions, starting from the base alloy, with gradually and systematically increased Cr contents. Their data clearly show that Cr addition significantly improves the corrosion resistance of the base amorphous alloy while only slightly changing its thermal and magnetic properties. In particular, the improvement on the corrosion resistance is highly impressive, bringing down the anodic current density by more than two orders of magnitude.
The manuscript is overall well written and organized. I enjoyed reading it. I suggest publication after the authors consider the few minor/optional changes listed below:
- The last sentence in the Abstract, in the version I received for review, seems to be incomplete, missing some words at the end. The end that I’m seeing is “through a variation of”. Please check on this.
- Section 2 (Materials & Methods), 1st paragraph, third-last sentence: “tnesile” should be “tensile”?
- Paragraph following Table 1, last sentence: “Thus we can be speculated …” might be better if changed to “Thus it can be speculated …”
- In the description of thermomechanical test results, “dimension charge” was used at several places (text, Figure 3 y-label and caption). I’m guessing the authors meant to say “dimension change” instead.
- In Fig. 6 caption, “… measured in Figure 4” might be “… measured in Figure 5”.
- In Conclusion: the last sentence in the 2nd paragraph is almost the same as the first sentence in the 3rd paragraph. Please consider revision.
Author Response

(The authors gave the same response as above.)

Reviewer 4 Report
The manuscript addresses the production and characterization of Fe-based metallic glass with the addition of Cr and evaluates such influence on (micro)structural features, magnetic properties, and anti-corrosion performance. Comments are given below:
ABSTRACT
- It is incomplete; the last sentence is not finished, so that it is unclear authors' comments/suggestions.
INTRODUCTION
- The term "relatively low cost" is something to be careful about. If compared with electrical steels, the statement is not accurate.
- What does "increase soft magnetic properties" mean? I understand reduce coercivity, so it might be misleading.
METHODS
- Clearly described.
RESULTS
- XRD, DSC: typical results for amorphous magnetic materials (Table 1 you don't need to add two decimal places for T because you might not have such measurement precision;
- I believe the word "charge" should read "change".
- Again the concept of high and low depends on the baseline: saturation polarization of 1.5T is not necessarily high for a soft magnetic material (electrical steel achieve about 1.9T); therefore, some specifics in terms of comparison basis is relevant. In addition, insets in Fig. 4 could be added in order to demonstrate Hc values for the alloys (in the presented scale, all looks equal to zero).
Author Response

(The authors gave the same response as above.)

Round 2
Reviewer 4 Report
The authors included proposed suggestions.